# Effects of Dietary *Bacillus amyloliquefaciens* SCAU-070 (Based on a Woody Plant-Based Diet) on Antioxidation, Immune and Intestinal Microbiota of Tilapia (*Oreochromis niloticus*)

**DOI:** 10.3390/microorganisms12061049

**Published:** 2024-05-23

**Authors:** Qijing Chen, Feng Wu, Xinye Chen, Qiaoting Yang, Biyin Ye, Xiaoyu Chen, Xiaoyong Zhang, Qin Pan

**Affiliations:** University Joint Laboratory of Guangdong Province, Hong Kong and Macao Region on Marine Bioresource Conservation and Exploitation, College of Marine Sciences, South China Agricultural University, Guangzhou 510642, China; chenqijing77@163.com (Q.C.); wufeng4505@163.com (F.W.); xinyeeast@stu.scau.edu.cn (X.C.); 13480325293@163.com (Q.Y.); 13640009806@163.com (X.C.)

**Keywords:** *Bacillus amyloliquefaciens*, intestinal morphology, antioxidation, non-specific immunity, microbiota

## Abstract

This study aimed to explore the effects of *Bacillus amyloliquefaciens* (BA) as one woody forage addition (as a probiotic, 1 × 10^7^ CFU/g) on tilapia (*Oreochromis niloticus*). Woody forage is one kind of fishery feed that could significantly enhance the growth, feed utilization, and digestibility of tilapia. At first, tilapia was divided into eight groups and fed with control, control + BA, *Moringa oleifera*, *M. oleifera* + BA, *Neolamarckia cadamba*, *N. cadamba* + BA, *Broussonetia papyrifera*, and *B. papyrifera* + BA diets, respectively. After dieting for 8 weeks, the intestinal morphology of tilapia in the eight groups was observed, and the effects of the *B. amyloliquefaciens* addition and wordy forage on the intestine functions were analyzed by two-way ANOVA. As no significant negative effects were found on the woody forage on tilapia, the villus height, density and width, and epithelial goblet cells in the posterior intestines of tilapia with BA supplementation were greater than those in the groups without BA supplementation, suggesting *B. amyloliquefaciens* SCAU-070 could promote the growth and development of tilapia intestinal tracts. Furthermore, it was found that *B. amyloliquefaciens* SCAU-070 enhanced the antioxidation capacity of tilapia posterior intestine tissue by promoting the activity of superoxide dismutase and content of malondialdehyde. In addition, the result of high-throughput sequencing (16S rDNA) showed that the beneficial bacteria *Cetobacterium* and *Romboutsia* in the probiotic groups increased significantly, while the potential pathogenic bacteria *Acinetobacter* decreased significantly.

## 1. Introduction

Tilapia, as second only to carp in global edible fish production and one of the most widely reproduced and adaptable fish in the world among many aquaculture species, has contributed greatly to global food protein due to their relatively cheap cost and high-quality protein [1]. Currently, the tilapia culture mode has gradually changed from the traditional semi-intensive culture to the high-density, intensive, and factory artificial culture. However, such a factory culturing mode causes serious pollution to the rearing water environments, leading to decreased immunity in tilapia and outbreaks of diseases [2]. Since the 1950s, antibiotics have been widely used in diets to prevent aquatic diseases and increase production, leading to the pollution of antibiotics in the aquaculture water environments, and the antibiotics remaining in animals will eventually endanger human health [3]. Therefore, it is particularly important to find safe substitutes for antibiotics to ensure the sustainable and healthy development of aquaculture. At present, the application of probiotics in aquaculture has attracted widespread attention because of its safety, low pollution, and other advantages [4,5,6].

Probiotics can generate health effects by improving the microecological balance of the host, as well as playing the role of the intestinal tract more effectively [7]. With the demand of environment-friendly agriculture, probiotics have been used to improve the growth of aquatic animals, diet utilization rate, natural immunity, and disease resistance [8]. According to previous studies, it has been found that various types of probiotics can increase the feed utilization, growth, and infectious disease resistance of tilapia [9]. In the process of culturing *Channa striata*, the diet with probiotics led to better growth performance, nutrient digestibility, and the expression of immune regulatory genes compared to the control diet without probiotics [10]. In addition, probiotics are also considered as a strategy to improve plant protein digestion to replace fish meals [11].

In recent years, the genus *Bacillus* has been widely used in aquaculture [12,13], and it may produce a variety of secondary metabolites with antibacterial and antifungal activities, including small peptides and proteins, polyketones, and lipopeptides [14,15,16]. For example, *Bacillus licheniformis* and *B. flexus* can effectively enhance the growth, digestive enzyme activities, innate immune enzyme activities, stress tolerance, and disease resistance of *Litopenaeus vannamei* [17]. As an important member of the genus *Bacillus*, the applications of *B. amyloliquefaciens* in aquaculture have been investigated in the past decade [18].

The positive effects of three woody forages, *Moringa oleifera*, *Neolamarckia cadamba*, and *Broussonetia papyrifera*, as foods supplementary to the growth, feed utilization, apparent digestibility, intestinal morphology, and microbiota composition of tilapia, have been validated in a previous study [19]. In order to make better use of these woody feeds, *B. amyloliquefaciens* SCAU-070, isolated from the intestinal tract of tilapia, was added to woody feeds in this study, and the effects of *B. amyloliquefaciens* on the intestinal morphology, immune-related gene expression, and intestinal microbiota of the tilapia were investigated by traditional methods and high-throughput sequencing. These results might be great help in the further exploitation of new diets (woody forages) and probiotic sources in aquaculture.

## 2. Materials and Methods

### 2.1. Strain B. amyloliquefaciens

The strain isolated from the intestines of tilapia, was cultured according to a previous method [18]. In brief, *B. amyloliquefaciens* was inoculated in seed liquid medium (peptone 10 g, NaCl 5 g, beef extract 5 g, glucose 10 g, yeast extract 5 g, and distilled water 1000 mL, pH 7) for 24 h at 30 °C and 160 r/min. The final concentration was adjusted to 10^10^ cells/mL according to a blood cell counter. Subsequently, the bacterial suspension was centrifuged at 5000 r/min for 10 min, the supernatant was removed, and the cells were washed twice with PBS; after that, they were resuspended in PBS and freeze-dried. A total of 12 g dry powder of *B. amyloliquefaciens* (approximately 10^11^ CFU/g) was obtained.

### 2.2. Test Diet Preparation

Eight kinds of diets were defined as the control (CK), control + *B. amyloliquefaciens* (CK + BA), *Moringa oleifera* group (MOL), *M. oleifera* + *B. amyloliquefaciens* (MOL + BA), *Neolamarckia cadamba* group (NC), *N. cadamba* + *B. amyloliquefaciens* (NC + BA), *B. papyrifera group* (BP), and *B. papyrifera* + *B. amyloliquefaciens* (BP + BA), respectively. All feeds were formulated to be isoprotein and isoenergetic. The basic components of the diet in each group are listed in Appendix A. The content of the woody diet added in each group was the best content obtained in the pre-experiment and kept the relative consistency of crude protein. Dry powder of *B. amyloliquefaciens* (final concentration was 1 × 10^7^ CFU/g diet) was added into the diets of groups CK + BA, MOL + BA, NC + BA, and BP + BA, respectively.

### 2.3. Feeding Experiment

Tilapia fry in the experiment were obtained from the Tilapia Breeding Farm in Guangzhou, China, and the test site was in the indoor circulating water culture system of the College of Marine Sciences, South China Agricultural University. Before the start of the formal culture trial, the mixed diet (four diets without *B. amyloliquefaciens* were mixed) was supplied and domesticated for 2 weeks, so as to adapt to the environmental conditions. Subsequently, the fish (11.3 ± 0.9 g) were divided into 24 culture tanks, with 30 fish in each culture tank, and tilapia in every 3 culture tanks ate the same group diet. Generally, younger or smaller fish often require a higher percentage of their body weight in feed compared to larger or mature fish, because their growth rate is higher. Hence, in our experiment, we fed them 8% of fish body weights with fish weights < 20 g, 7% of fish body weights with fish 20 ≤ weights < 30 g, 6% of fish body weights with fish 30 ≤ weights < 40 g, and 5% of fish body weights with fish weights ≥ 40 g. The experimental diet was fed to the fish 3 times a day for 56 days. The water environment parameters were as follows: the pH value was 7.6–7.8, the temperature was 28–32 °C, the dissolved oxygen content was greater than 4 mg/L, and the ammonia nitrogen content was less than 0.1 mg/L.

### 2.4. Sample Collection

This experiment was conducted under the guidance of the Animal Ethics Committee (approval ID: SYXK-2019-0136). When the feeding trial was finished, feed was withheld from the fish for 24 h. Additionally, two fish from each treatment group were anesthetized by tricaine methane sulphonate (MS 222, 100 ppm), so as to collect the gut samples and stored in a 2 mL cryopreservation tube. In addition, four fish were randomly selected from each group for euthanasia. Subsequently, the intestines were collected aseptically from the abdominal cavity, and the contents of the posterior intestines of each group of four fish were collected into sterile tubes, which were quickly frozen under liquid nitrogen and then stored at −80 °C. We obtained samples from two distinct sets of gut samples and four sets of intestinal contents.

### 2.5. Observation of Posterior Intestine Tissue Sections

The posterior intestine (1–2 cm) tissue was gently washed with 0.86% normal saline, then fixed with Bouin’s solution. After 24 h, it was transferred to 70% alcohol for the paraffin sections. During the process of making the sections, the tissue sample was subjected to a routine series of gradient ethanol dehydration, dimethylbenzene transparency, paraffin embedding, sections (5 μm), and H&E staining and then observed and photographed. Meanwhile, the villus height (VH), villus width (VW), villus density (VD), and muscular thickness (MT) were measured under an ordinary optical microscope, and the number of goblet cells (GCs) was counted.

### 2.6. Determination of Antioxidation Indicators

The posterior intestine sample was placed in a centrifuge tube with a cold phosphate buffer (PBS, pH 7.4) and then homogenized in an ice bath. The sample was then subjected to 3500 r/min for 15 min, washed twice with cold PBS buffer (pH 7.4), and then suspended in PBS. The activity of superoxide dismutase (SOD (#A001-3-2 from Nanjing Jiancheng Bioengineering Institute, Nanjing, China)), the antioxidant capacity (T-AOC (#A015-2-1 from Nanjing Jiancheng Bioengineering Institute, Nanjing, China)), and the content of malondialdehyde (MDA (#A003-1-2 from Nanjing Jiancheng Bioengineering Institute, Nanjing, China)) in the supernatant were detected by a commercial kit according to the manufacturer’s instructions.

### 2.7. Expression Analysis of Immune-Related Genes

The effect of *B. amyloliquefaciens* SCAU-070 on the transcription of the intestinal immune-related genes was detected by using real-time quantitative polymerase chain reaction (RT-qPCR). Total RNA was extracted from the posterior intestine of tilapia by the Sinply P Total RNA Extraction Kit (BioFlux, Hangzhou, China). The concentration of total RNA was determined and analyzed by a nucleic acid analyzer, and then, the quality of the extracted RNA was detected by 1% agarose gel electrophoresis. The cDNA template was synthesized using the reverse transcription kit PrimeSript^TM^ RT Regent Kit (Takara, San Jose, CA, USA). Then, fluorescence quantitative PCR was performed using a QuantStudio 5 real-time fluorescence quantitative PCR gene amplification instrument (Life Technologies, Carlsbad, CA, USA) with THUNDERBIRD^®^ SYBR qPCR Mix (TOYOBO, Tokyo, Japan), and the reaction system was 10 μL. Primers for the immune-related genes are shown in Appendix A, including tumor necrosis factor-α (TNF-α), interleukin 6 (IL-6), interleukin 10 (IL-10), complement C3, and housekeeping gene β-actin, all designed with Premier 5.0. The PCR program was as follows: 1 min at 95 °C; 40 cycles (15 s at 95 °C, 15 s at 60 °C, and 45 s at 72 °C); and 5 min at 72 °C. The β-actin gene was used as the internal reference of the transcription level of the target gene, and the relative expression was determined by the 2^−ΔΔCT^ method.

### 2.8. Intestinal Microbial Diversity and Statistical Analysis

#### 2.8.1. DNA Extraction and Microbial 16S rDNA Amplicon Sequencing

DNA extraction, sample quality control, library construction, and sequencing were carried out by Genewiz Company (Suzhou, China). Briefly, the total microbial DNA was extracted from 32 samples using a DNA kit (Omega Bio-Tek, Norcross, GA, USA). The quantity and quality of the DNA were measured by a Quant-iT PicoGreen dsDNA analysis kit and NanoDrop Lite spectrophotometer (Thermo Scientific, Waltham, MA, USA), respectively.

Using 20–30 ng DNA as the template, a series of PCR primers designed by Genewiz Company (Suzhou, China) were used to amplify two highly variable regions of prokaryotic 16S rDNA, V3 and V4. The V3 and V4 regions were amplified using forward primers containing the sequence “CCTACGGRRBGCASCAGKVRVGAAT” and reverse primers containing the sequence “GGACTACNVGGGTWTCTAATCC”. In addition, an adapter with an indexing function was added to the end of the PCR product of 16S rDNA by PCR to sequence the NGS. The PCR reaction parameters were as follows: 3 min at 94 °C; 24 cycles (5 s at 94 °C, 90 s at 57 °C, and 10 s at 72 °C); and 5 min at 72 °C. PCR products were detected by 1.5% agarose gel electrophoresis.

#### 2.8.2. Microbial Operational Taxonomic Unit (OTU) Cluster and Taxonomic Annotation

The PCR product was detected by a microplate reader and sequenced at two ends according to Illumina MiSeq/NovaSeq (Illumina, San Diego, CA, USA), and the sequence information was read by the relevant software attached to the instrument. After quality filtering, the chimeric sequences were removed, and the final sequences were used for OTU clustering. Sequence clustering (sequence similarity was set to 97%) was carried out by using VSEARCH (1.9.6), and the 16S rDNA database was Silva 132. The representative sequences (of each OTU) were analyzed by using the RDP classifier (Ribosomal Database Program) Bayesian algorithm, and the community composition of each sample was calculated at different species classification levels.

According to the results of the OTU analysis, Shannon, Chao 1, and other α-diversity indices were calculated to reflect the species abundance and diversity of the community by random sampling of the sample sequences. The significant differences in the microbial community between samples were compared by (un)weighted UniFrac analysis, and the principal coordinate analysis (PCoA) was based on the Bray–Curtis distance matrix between samples to show β-diversity visualization [20]. Functional pathways were predicted using PICRUST and annotated using the Kyoto Encyclopedia of Genomics and Genomics (KEGG). Significant differences in functional pathways in the intestinal microbiota of tilapia from different diets were identified by *t*-test (*p* < 0.05), and all *p*-values were corrected for the false detection rate (FDR). The FDR-corrected *p* < 0.05 was considered significant.

The experimental results were expressed by the mean ± standard error (SE), and the data were analyzed by one-way analysis of variance (ANOVA) with SPSS 20.0 software. If a significant difference was found between groups, Duncan’s multiple comparison test was carried out; *p* < 0.05 indicated a significant difference. The impact levels of woody forages and *B. amyloliquefaciens* addition to tilapia, as well as their interactions, were evaluated by two-way ANOVA. When overall differences were detected, differences between the means were tested using Tukey’s multiple range test, and all differences were considered significant at *p* < 0.05.

## 3. Results

### 3.1. Histomorphological Structure

After feeding for 56 days, different diets had different effects on the intestinal morphology of tilapia (Table 1). Two-way ANOVA demonstrated that woody forages could significantly increase the number of goblet cells within the intestines of tilapia, while all intestinal morphological characteristics (VH, VM, VD, and goblet cells) except MT expressed significant increases with the supplementation of *B. amyloliquefaciens* SCAU-070 (Table 2). In addition, the number of goblet cells in the BP and MOL groups was significantly increased compared to the CK group. Under the microscope, the morphological structures of the intestinal mucosa of all groups of tilapia were relatively complete and well-defined, with dense and regularly arranged intestinal mucosal epithelial cells, clear contours, and distinct staining under high magnification (Figure 1). In addition, the VD, VW, and VH of the posterior intestine in the four probiotic groups of tilapia were observed to increase compared to the four non-probiotic groups in Figure 1. Moreover, the thickness of the villi (in the posterior intestine) also showed a significant change (Figure 1), i.e., the thickness of the muscle layer (in the posterior intestine) increased significantly in the four non-probiotic groups compared to the four probiotic groups, respectively.

### 3.2. Determination of the Antioxidant Index in the Posterior Intestine

The SOD and T-AOC activity in the three woody forage groups were significantly higher than that in the CK group (*p* < 0.05), and the *B. amyloliquefaciens* supplementary groups showed significant increases in SOD and T-AOC and decrease in MDA concentrations. Additionally, the SOD activity in the NC and MOL groups was significantly higher than the control group, as was the T-AOC activity in the MOL group. However, no significant synergetic or antagonistic effect was detected in the woody forage + *B. amyloliquefaciens* group (Table 3 and Table 4).

### 3.3. Expression of Posterior Intestine Immune-Related Genes

The relative repression of *tnf-α*, *il-6*, *il-10*, and *c3* in the woody forage groups showed no significant differences compared to the CK group, while, in the *B. amyloliquefaciens* SCAU-070 supplementary groups, these genes were significantly upregulated in the intestines of tilapia (Figure 2). However, there were also no significant interactions in the wood forage and *B. amyloliquefaciens* SCAU-070 supplementary groups.

### 3.4. Microbiota

A total of 3,778,700 high-quality sequencing readings were generated from 32 samples in eight groups. The number of operational taxonomic units (OTUs), coverage, statistical estimates of species richness, and diversity indices for each sample at a genetic distance of 3% are presented in Appendix A. These sequences were high-quality sequences with up to 97% identity and aggregated into 14,416 OTUs, each of which contained different phylogenetic OTUs (116–691). The abundance of microbiota was calculated by using the Shannon index and Simpson index. The ranges of the Shannon index and Simpson index were 0.90–6.15 and 0.18–0.96, respectively. The Chao index and ACE index were usually used to calculate the microbial diversity. The Chao index ranged from 203.40 to 818.44, while the ACE index ranged from 203.30 to 812.27. The coverage of each sample for estimating the sequencing integrity was 0.99, and all rarefaction curves (composed of the microbial OTU number versus sequence) from the eight groups of tilapia posterior intestines showed a saturation platform (Appendix A), indicating that the identified sequence can represent the vast majority of the bacteria in each sample.

All the OTUs were identified as 21 phyla from 16S rDNA sequencing. The composition at the phylum level in the microbiota of the posterior intestines of tilapia fed eight different diets is shown in Figure 3A. Among them, Fusobacteriota (10.92–66.44%), Bacillota (10.68–43.81%), Pseudomonadota (4.23–42.95%), Bacteroidota (1.01–21.80%), Actinomycetota (1.03–14.69%), and Chloroflexota (0.38–11.85%) were the six phyla with the highest relative abundance (>90%). The top 30 microbial compositions of each group are shown in Figure 3B. Among them, 14 of the top 30 genera were unclassified. In all eight groups, *Cetobacterium* and *Bacillus* were the dominant genera.

The PCoA based on the Bray–Curtis distance matrix was used to compare the similarity of the microbial community of 32 samples (Appendix A). According to the result of the PCoA, it showed that the samples were clustered according to the type of diets, and there was significant segregation between the CK and CK + BA groups, between the MOL and MOL + BA groups, between the NC and NC + BA groups, and between the BP and BP + BA groups (Appendix A), which indicated that the *B. amyloliquefaciens* SCAU-070 addition in the diet had a strong influence on the overall structure of the posterior intestine microbiota of the tilapia.

The KEGG pathways with obvious differences were divided into six categories based on level: “metabolism”, “genetic information processing”, “cell processing”, “environmental information processing”, “organic system”, and “human disease” (Figure 4). It is worth noting that, compared to the CK group, the KEGG pathways related to carbohydrate metabolism, cell growth, and death increased significantly in the CK + BA group (Figure 5A). The KEGG pathways associated with translation, replication and repair, the endocrine system, folding classification and degradation, the immune system, immune system diseases, environmental adaptation, nucleotide metabolism, the biosynthesis of other secondary metabolites, signal molecules and interactions, and carbohydrate metabolism in the MOL + BA group were significantly higher than those in the MOL group (Figure 5B). Compared to the BP group, the KEGG pathways related to environmental adaptability, metabolism, cellular process and signal transduction, nucleotide metabolism, genetic information processing, translation, the immune system, signal molecules and interactions, and glycan biosynthesis metabolism increased significantly in the BP + BA group (Figure 5C).

## 4. Discussion

As the largest digestive and immune organ in animals, the intestinal tract plays a crucial part in regulating the nutrition and immunity of animals [21]. Histomorphological indices such as the intestinal VH, VD, VW, MT, and goblet cell number are often used to reflect the development and health of fish intestinal tracts and to evaluate their digestion and absorption, antioxidation, and immune capacity [22,23]. Our results in this study showed that, although no significant differences were detected in the morphology, antioxidant capacity, and immune-related gene expressions of tilapia intestines between the CK and three woody forage groups (Figure 1, Table 1 and Table 2), the addition of *B. amyloliquefaciens* SCAU-070 in all diets significantly promoted the intestinal parameters, including VH, VD, and VW, which might be related to the regulation of intestinal microbiota by *B. amyloliquefaciens* [24]. In contrast, the MT of tilapia intestines was significantly decreased with the addition of *B. amyloliquefaciens* SCAU-070, indicating that tilapia may reduce the energy needed to maintain the basic intestinal physiological activity, so that such energy can be used for growth or the absorption of nutrients.

The results in this study also showed that the number of goblet cells in the posterior intestine epithelial layer of the tilapia for the *B. amyloliquefaciens* SCAU-070 diet were significantly higher, meaning that adding *B. amyloliquefaciens* SCAU-070 into the diet was beneficial to the occurrence of goblet cells in the fish intestinal epithelium. In addition, although no significant differences were observed in the morphology, antioxidant capacity, and immune-related gene expressions of tilapia intestines between the control group (CK) and the woody forage groups (Figure 1, Table 1 and Table 2), the *B. amyloliquefaciens* SCAU-070 supplementation could significantly enhanced these intestinal parameters. The most important function of the goblet cells was to secrete mucus and mucin, which played a crucial part in protecting the epithelium. Furthermore, mucin could prevent the invasion of pathogens, and its existence might serve as a particularly important innate defense barrier [25]. At the same time, mucus could lubricate the epithelial surface and promote defecation in the posterior intestine. Therefore, the addition of *B. amyloliquefaciens* SCAU-070 to diets could effectively protect the intestinal health of tilapia from pathogens.

It was very important to explore the antioxidation capacity of the tilapia intestinal tract. After intestinal oxidative damage, pathogenic bacteria easily invade fish [26]. In this experiment, the SOD activity of tilapia in the *B. amyloliquefaciens* SCAU-070 supplementation groups were improved compared to those without *B. amyloliquefaciens* (Table 3 and Table 4). T-AOC reflected the total antioxidation defense level of the body [27]. In the same way, the T-AOC activity of tilapia fed with *B. amyloliquefaciens* SCAU-070 was also improved compared to those without *B. amyloliquefaciens*. The content of MDA can reflect the degree of membrane lipid peroxidation and can also indirectly reflect the strength of the antioxidation effect [28], and the MDA content was also reduced compared to those without *B. amyloliquefaciens*. These measured indices were closely related to the antioxidation function of the body. This indicated that *B. amyloliquefaciens* SCAU-070 enhanced the antioxidation capacity of tilapia posterior intestine tissue.

Cytokines played a crucial part in the immune system. Among them, proinflammatory cytokines such as *TNF*-α and IL-6 are commonly used reference genes in immune regulation research [29]. Previous studies have shown that supplementing *B. amyloliquefaciens* into diets can change the expression level of proinflammatory cytokines in fish [30]. In this study, the expression of TNF-α gene in the *B. amyloliquefaciens* SCAU-070 supplementary groups were significantly upregulated. Moreover, the expression level of the IL-6 gene in the four groups with the addition of *B. amyloliquefaciens* SCAU-070 was upregulated compared to that in the four groups without *B. amyloliquefaciens*. The changes of these gene expression levels might be influenced by the intestinal microbiota [29]. Anti-inflammatory cytokines can limit the activity of pathogens and prevent damage to host tissues and maintain immune homeostasis [31]. IL-10, an anti-inflammatory cytokine, is often used in the study of immune regulation mechanisms [31]. The expression level of the IL-10 gene in the four diet groups with *B. amyloliquefaciens* SCAU-070 added was significantly upregulated compared to that in the four diet groups without *B. amyloliquefaciens* (*p* < 0.05), which indicated that tilapia could obtain a better immune balance after adding *B. amyloliquefaciens*. Complement C3 can mediate the immune response and inflammatory response and plays a crucial part in both classical activation pathways and bypass activation pathways of complements [32,33]. Compared to the four non-probiotic groups, the expression level of the C3 gene in the four probiotic groups was significantly upregulated (*p* < 0.05). The increase in the C3 gene expression level indicated that the tilapia could activate the complement system faster when fed with *B. amyloliquefaciens*.

In this study, Illumina sequencing technology was used to study the composition of the intestinal microorganisms in tilapia. In all the eight diet groups, the six phyla with the highest abundance were Fusobacteriota, Bacillota, Pseudomonadota, Bacteroidota, Actinomycetota, and Chloroflexota (Figure 3). Among them, the first five bacterial phyla can be widely distributed in tilapia intestines [34], while Chloroflexota was firstly discovered in the intestines of tilapia in this study. Further comparisons revealed that all the eight diet groups except for the CK and NC groups had a relatively high proportion of Chloroflexota, indicating that dietary woody feed and *B. amyloliquefaciens* SCAU-070 might affect the microbial community in the intestines of tilapia. At the genus level, *Cetobacterium* and *Bacillus* were the dominant bacteria in the intestines of tilapia, consistent with previous studies [35]. However, it was not difficult to find that the abundance of some genera of the tilapia posterior intestinal microbiota in the four probiotic groups were significantly downregulated (Figure 3A). Among them, *Acinetobacter* was related to pathogenic bacteria that can cause the sudden death of mandarin fish (*Siniperca chuatsi*) [36]. In addition, compared to the CK, MOL, and BP groups, the abundance of *Cetobacterium* in the CK + BA, MOL + BA and BP + BA groups was significantly higher. As a potentially beneficial bacterium, *Cetobacterium* is usually found in the intestinal tract of freshwater fish [37] and can produce vitamin B12 [36] and acetic acid and promote the metabolism of proteins, carbohydrates, and fat [38].

*Romboutsia* is a species that can utilize a series of carbohydrates in different ways [38], and the abundance of this potentially beneficial bacterium was significantly upregulated in the four probiotic groups (Figure 3). At the same time, this is a genus that has never been reported in fish intestines. Xia et al. (2020) found that *B. amyloliquefaciens* could affect the composition of intestinal microbiota [39], thus changing the intestinal metabolism and, finally, led to the improvement of host growth and immunity [40]. KEGG function analysis of the microbial community in this experiment (Figure 4 and Figure 5) showed that the metabolic pathways were obviously different between the probiotic groups and non-probiotic groups. Compared to the CK group, the KEGG pathways associated with carbohydrate metabolism cell growth and death in the CK + BA group increased significantly. The KEGG pathways related to translation replication and repair endocrine system, folding classification, degradation immune system diseases, environmental adaptation, nucleotide metabolism, the biosynthesis of other secondary metabolites, signal molecules and interactions, and carbohydrate metabolism in the MOL + BA group were significantly higher than those in the MOL group. The KEGG pathways associated with environmental adaptability, metabolism, cellular processes and signal transduction, nucleotide metabolism, genetic information processing, translation, the immune system, signal molecules and interactions, and glycan biosynthesis and metabolism in the BP + BA group were significantly higher than those in the BP group. This is consistent with our previous results: *B. amyloliquefaciens* can promote the growth of tilapia and improve the immunity of tilapia. Therefore, adding *B. amyloliquefaciens* SCAU-070 to tilapia feed could improve the health rate of tilapia in aquaculture.

## 5. Conclusions

Overall, this is the first investigation on the effects of dietary *B. amyloliquefaciens* on the antioxidation, immunity, and intestinal microbiota of tilapia fed with woody diets. It was helpful to better understand the health of cultured tilapia by studying the effects of the diets on the intestinal tract. While no adverse effects were found in the woody forage diets on tilapia in the intestinal histomorphological structure, antioxidative capacity, and immune gene expression, the addition of *B. amyloliquefaciens* SCAU-070 could significantly affect the intestine structure and morphology, antioxidative capacity, and immunity in a positive way, irrespective of the feed type. Moreover, no specific synergy or antagonism between *B. amyloliquefaciens* SCAU-070 and the woody forage were found. However, more studies are needed to better explain the mechanisms of the interactions between the intestinal tract and microorganisms. Therefore, the next phase of research should focus on the changes in the intestinal microorganisms at different growth stages and dieting periods, so as to determine the change processes of intestinal microorganisms.

## Figures and Tables

**Figure 1 microorganisms-12-01049-f001:**
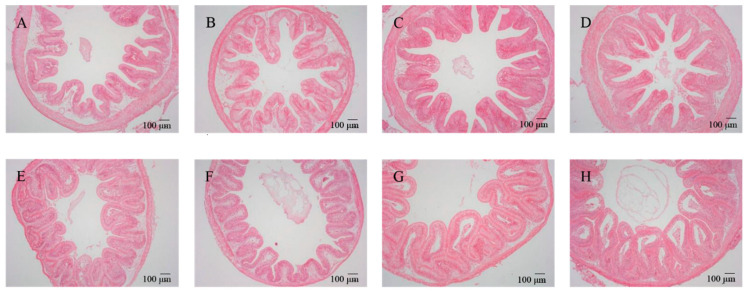
Effects of dietary *B. amyloliquefaciens* on the distal intestine morphology and structure of tilapia (×100). (**A**) CK group, control diet; (**B**) MOL group, *Moringa oleifera* diet; (**C**) NC group, *Neolamarckia cadamba* diet; (**D**) BP group, *Broussonetia papyrifera* diet; (**E**) CK + BA group, control + *Bacillus amyloliquefaciens* (BA) diet; (**F**) MOL + BA group, *M. oleifera* + BA diet; (**G**) NC + BA group, *N. cadamba* + BA diet; (**H**) BP + BA group, *B. papyrifera* + BA diet.

**Figure 2 microorganisms-12-01049-f002:**
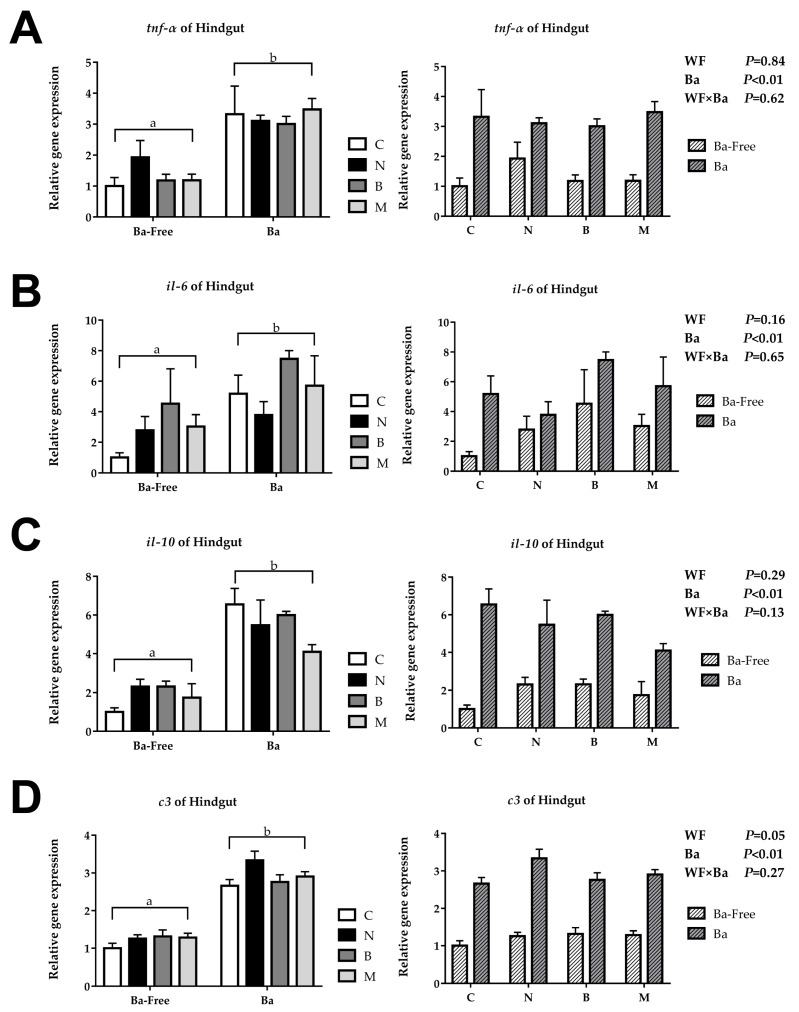
Immune-related gene expression of tilapia posterior intestine: (**A**) tumor necrosis factor-α (*tnf*-α), (**B**) interleukin 6 (*il*-6), (**C**) interleukin 10 (*il*-10), and (**D**) complement C3 (*c3*). C, control diet; M, *Moringa oleifera* diet; N, *Neolamarckia cadamba* diet; B, *Broussonetia papyrifera* diet; Ba, supplementation of *Bacillus amyloliquefaciens* SCAU-070. Different lowercase letters (a, b) indicate significant differences (*p* < 0.05) between the *B. amyloliquefaciens* addition and non-*B. amyloliquefaciens* addition groups.

**Figure 3 microorganisms-12-01049-f003:**
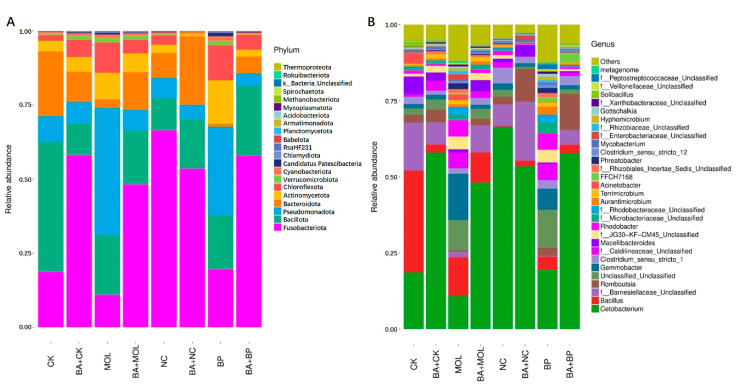
Microbiota composition of the bacterial taxa at the phylum level in the eight groups (**A**). Microbiota composition of the bacterial taxa at the genus level in the eight groups (**B**). CK, control diet; CK + BA, control + *Bacillus amyloliquefaciens* (BA) diet; MOL, *Moringa oleifera* diet; MOL + BA, *M. oleifera* + BA diet; NC, *Neolamarckia cadamba* diet; NC + BA, *N. cadamba* + BA diet; BP, *Broussonetia papyrifera* diet; BP + BA, *B. papyrifera* + BA diet.

**Figure 4 microorganisms-12-01049-f004:**
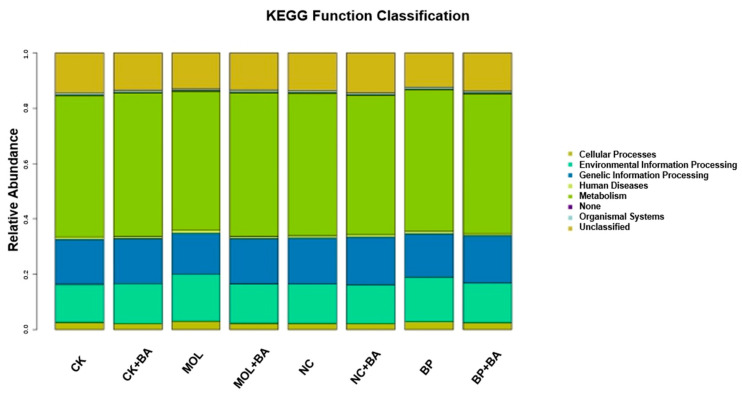
Column distribution of the KEGG function abundance in each group. CK, control diet; CK + BA, control + *Bacillus amyloliquefaciens* (BA) diet; MOL, *Moringa oleifera* diet; MOL + BA, *M. oleifera* + BA diet; NC, *Neolamarckia cadamba* diet; NC + BA, *N. cadamba* + BA diet; BP, *Broussonetia papyrifera* diet; BP + BA, *B. papyrifera* + BA diet.

**Figure 5 microorganisms-12-01049-f005:**
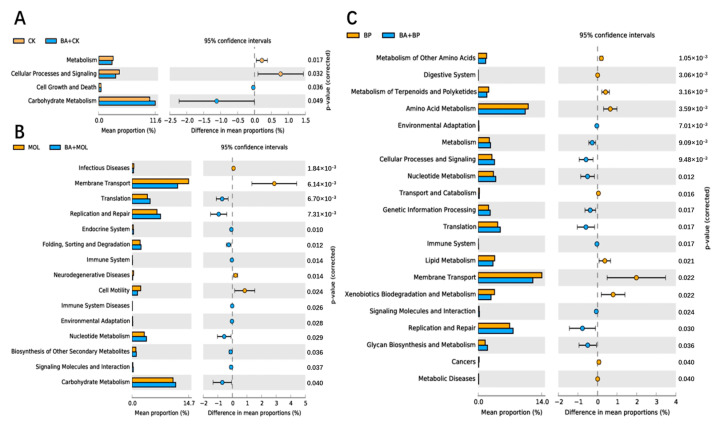
Differential analysis on the KEGG pathways of intestinal microbes: (**A**) groups CK (control diet) and CK + BA (control + *Bacillus amyloliquefaciens* (BA) diet); (**B**) groups MOL (*Moringa oleifera* diet) and MOL + BA (*M. oleifera* + BA diet); (**C**) groups BP (*Broussonetia papyrifera* diet) and BP + BA (*B. papyrifera* + BA diet).

**Table 1 microorganisms-12-01049-t001:** Effects of *B. amyloliquefaciens* added to different diets on the intestinal morphology of tilapia.

	CK	CK + BA	MOL	MOL + BA	NC	NC + BA	BP	BP + BA
VH (μm)	264.85 ± 37.05	305.75 ± 9.42	266.21 ± 26.10	335.35 ± 11.42	275.11 ± 9.16	292.70 ± 9.35	278.42 ± 12.75	302.08 ± 10.28
VW (μm)	88.42 ± 6.61	103.84 ± 7.80	90.82 ± 3.89	103.92 ± 16.22	99.87 ± 7.58	112.39 ± 12.63	90.10 ± 4.91	109.81 ± 7.38
Muscular thickness (μm)	62.24 ± 1.51	44.62 ± 2.77	69.52 ± 2.33	51.36 ± 1.05	66.48 ± 1.86	45.09 ± 3.35	78.79 ± 1.42	53.70 ± 2.58
VD	19.00 ± 5.00	23.33 ± 3.33	22.67 ± 3.71	22.67 ± 2.73	16.67 ± 1.86	17.00 ± 1.15	20.33 ± 2.40	26.00 ± 0.58
goblet cells	14.67 ± 1.67	16.33 ± 0.88	8.00 ± 0.58	19.00 ± 0.58	5.33 ± 1.20	8.67 ± 0.88	9.67 ± 0.33	18.33 ± 1.76

Values are CK, control diet; CK + BA, control + *Bacillus amyloliquefaciens* (BA) diet; MOL, *Moringa oleifera* diet; MOL + BA, *M. oleifera* + BA diet; NC, *Neolamarckia cadamba* diet; NC + BA, *N. cadamba* + BA diet; BP, *Broussonetia papyrifera* diet; BP + BA, *B. papyrifera* + BA diet.

**Table 2 microorganisms-12-01049-t002:** Effects of woody forages, *B. amyloliquefaciens*, and their interaction on the intestinal morphology of tilapia through two-way ANVOA.

	VH (μm)	VW (μm)	VD	Muscular Thickness (μm)	Goblet Cells
Woody forage	0.793	0.709	0.071	0.837	0.001 (a/ab/bc/c)
*B. amyloliquefaciens*	0.010 (x/y)	0.033 (x/y)	0.000 (x/y)	0.004 (y/x)	0.000 (x/y)
Woody forage × *B. amyloliquefaciens*	0.519	0.979	0.949	0.984	0.783

The lowercase letters in parentheses following the *p*-values indicate significant differences, as determined by two-factor ANOVA. Different lowercase letters indicate significant differences (*p* < 0.05) among different woody forages or between the *B. amyloliquefaciens* addition and non-*B. amyloliquefaciens* addition groups. Each set of letters, for example, in parentheses (a/ab/bc/c) corresponds to groups CK, NC, BP, and MOL, respectively.

**Table 3 microorganisms-12-01049-t003:** Effects of *B. amyloliquefaciens* added to different diets on the intestinal antioxidant capacity of tilapia.

Index	CK	CK + BA	MOL	MOL + BA	NC	NC + BA	BP	BP + BA
SOD (U/mgprot)	13.84 ± 1.09	17.20 ± 2.30	19.83 ± 1.46	28.84 ± 3.84	18.63 ± 0.39	24.01 ± 1.66	17.98 ± 0.58	20.16 ± 1.94
T-AOC (mmol/gprot)	1.63 ± 0.08	1.95 ± 0.13	1.70 ± 0.09	1.98 ± 0.07	2.04 ± 0.16	2.04 ± 0.11	1.60 ± 0.09	1.92 ± 0.06
MDA (nmol/mgprot)	27.45 ± 1.53	23.83 ± 1.71	26.00 ± 1.61	22.52 ± 1.07	25.55 ± 1.81	23.39 ± 1.09	24.40 ± 1.85	22.94 ± 1.13

Values are means ± SD (n = 6). SOD = superoxide dismutase, T-AOC = total antioxidant capacity, MDA = malondialdehyde. CK, control diet; CK + BA, control + *Bacillus amyloliquefaciens* (BA) diet; MOL, *Moringa oleifera* diet; MOL + BA, *M. oleifera* + BA diet; NC, *Neolamarckia cadamba* diet; NC + BA, *N. cadamba* + BA diet; BP, *Broussonetia papyrifera* diet; BP + BA, *B. papyrifera* + BA diet.

**Table 4 microorganisms-12-01049-t004:** Effects of woody forages, *B. amyloliquefaciens*, and their interaction on the intestinal antioxidant capacity of tilapia through two-way ANVOA.

	SOD (U/mg prot)	T-AOC (mmol/g prot)	MDA (nmol/mg prot)
Woody forage	0.001 (a/bc/ab/c)	0.042 (a/b/a/ab)	0.620
*B. amyloliquefaciens*	0.009 (x/y)	0.003 (x/y)	0.016 (y/x)
Woody forage × *B. amyloliquefaciens*	0.376	0.354	0.867

The lowercase letters in parentheses following the *p*-values indicate significant differences, as determined by two-factor ANOVA. Different lowercase letters indicate significant differences (*p* < 0.05) among different woody forages or between the *B. amyloliquefaciens* addition and non-*B. amyloliquefaciens* addition groups. Each set of letters, for example, in parentheses (a/ab/bc/c) corresponds to groups CK, NC, BP, and MOL, respectively.

## Data Availability

The raw data supporting the conclusions of this article will be made available by the authors on request.

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
