# Peer review of "Effects of Dietary *Bacillus amyloliquefaciens* SCAU-070 (Based on a Woody Plant-Based Diet) on Antioxidation, Immune and Intestinal Microbiota of Tilapia (*Oreochromis niloticus*)"

_microorganisms, 2024, doi:10.3390/microorganisms12061049_

Round 1
Reviewer 1 Report
Comments and Suggestions for Authors
The present study evaluated the effects of dietary probiotics Bacillus amyloliquefaciens on antioxidation, immune and gut microbiota of Nile tilapia. Although the present study finished partial research works, however several questions are still needed to be improved. Thus, we think quality of manuscript remain insufficient to publish this eminent Journal based on following comments.
Specific comment:
1.Introduction line 60
The content mentioned that as an important member of genus Bacillus, B. amyloliquefaciens is rarely used in the cultivation of tilapia. However, if you survey the keyword B. amyloliquefaciens and tilapia in PubMed of NCBI, some papers had conducted B. amyloliquefaciens in tilapia relevant to the present study are not cited and discussed. We highly suggested the Introduction should be reorganized and supplemented the references in Introduction or Discussion part.
2. Line 62.
Authors mentioned that three different woody feeds had been applied in tilapia aquaculture and had achieved a good result (data not shown). However, the effect of three woody material on intestinal histomorphology compared to control seem no benefits based on the result showed in Table 1. For example, the goblet cells in three woody feed groups are significant reduced compared to that in CK group. Authors should strength the reasons in Introduction part why use three woody materials in feed and combine probiotic Bacillus amyloliquefaciens in feed.
3. Material and methods section 2.3 (Line 98)
The experimental diet was fed according to 5%-8% of fish body weight. Why not feed in a certain ratio of body weight? The protocol should be described in detail.
4. Result 3.1. Histomorphological structure
The results mentioned that the VD, VW and VH of posterior intestine in the four probiotic groups of tilapia were observed to increase compared with the four non-probiotic groups. The results whether reflected in improving of feed efficiency and growth performance. Unfortunately, the present study did not relevant date. We think these data are essential to support histomorphological results.
5. Result 3.3. Expression of posterior intestine immune related genes
The relative expression of TNF-a and IL-10 were significantly in the four groups with B. amyloliquefaciens (Figure 2). However, the balance between TNF-a and IL-10 is important for immune homeostasis maintenance. As discussion (line 362-364) mentioned that inflammatory cytokines could limit the activity of pathogens, and prevent damage to host tissues and maintain immune homeostasis, but the result showed in the study without pathogen infection. Both genes are up-regulated at the same time, the results appear to be contradictory. How to explain the result?
6. Although the expression of cytokine genes can be used as indicator of immune status, the resistance of pathogen infection is real effectiveness for immunomodulation. Thus, the data about the survival rate of fish in each group after challenging with pathogen presented in the study is high recommended.
General comments
1. Legend must be provided in Table 1.
2. Data in each list of Table 1 & 2 should have statistical analysis.
3. Figure 3 & Figure 5 are illegibility and difficult to read.
4. Bacterial concentration should be expressed as CFU/g
5. The strain number (SCAU-070) of B. amyloliquefaciens should be supplemented in all content.
Reviewer 2 Report
Comments and Suggestions for Authors
Major comments:
The authors examine the impact of eight different diets on tilapia. The experimental design is sufficient to assess the effects of these diets on the analyzed parameters.
Given that tilapia is cultivated for human consumption, it was anticipated that the average weights of the various groups would be provided. The authors conclude that fish fed with Bacillus amyloliquefaciens SCAU-070 in their diet are healthier. However, it remains unclear whether they are more profitable to sell compared to fish in the control group.
Characterizing one strain with probiotic features does not imply that all strains from that species possess similar features. This study focuses on the strain Bacillus amyloliquefaciens SCAU-070, and conclusions about other strains of the same species cannot be extrapolated until tested. Therefore, whenever referring to the species Bacillus amyloliquefaciens as a probiotic, the designation SCAU-070 should be included. This applies not only to the strain analyzed in this study but also when discussing probiotic strains from other research.
The definition of prebiotics incorporates selectively fermented ingredients that induce specific changes in the composition and/or activity of the gastrointestinal microbiota, thereby conferring benefits to the host's well-being and health. Typically, woody diets are considered potential prebiotics. In six assays, a woody diet is employed, while in two cases, it appears to be tested as a synbiotic (where prebiotics are added to probiotic foods, with their concentration in the product being below 10%). I suggest that the paper would benefit from analyzing the results within the framework of prebiotics and synbiotics concepts.
Specific comments:
Change microflora (8 times) to microbiota.
Change flora (3 times) to microbiota.
Line 12 – Correct the number of cells/g
Line 44 – “As a kind of active microorganisms beneficial to the host”. What do you mean by “as a kind”? A definition of a probiotic strain states that the consumption of the strain must result in a benefit to the host.
Lines 51, – Apply the italic format to the genus and species name.
Line 73 – Wu et al. (2020). Present the number of this reference.
Line 75 – “The final concentration”. Do you mean the initial concentration?
Lines 76-78 – Consider at least three changes in the next sentence: “Subsequently, the bacterial solution was centrifuged at 5000 r/min for 10 min, the supernatant was removed and the precipitation was washed twice with PBS, and finally PBS was added again and freeze-dried.”
- In “bacterial solution” change solution to suspension. Bacteria do not dissolve, representing a suspension and not a solution.
- The cells do not precipitate, they separate due to the higher density than the water.
Subsequently, the bacterial suspension was centrifuged at 5000 r/min for 10 min, the supernatant was removed and the cells were washed twice with PBS, after what were resuspended in PBS and freeze-dried.
Lines 196-197 – “Compared with the four probiotic groups, the muscular thickness of the four non-probiotic groups decreased significantly (10.75% ~ 26.34%).”
Please check the sentence since, according to data in Table 1, the probiotic groups have less muscular thickness values.
Lines 247-249 – Since 2021 these names are no longer valid. Check the literature or LPSN - List of Prokaryotic names with Standing in Nomenclature for the correct names and change them in accordance. Correct the names in the Figures also.
Figure 5 – Please use the same color for Control and Probiotic groups in Figures A, B and C.
References – Apply the italic format to the genus and species name in references 5, 8, 12, 14, 20, 28, and 36.
References – Apply the italic format to expression in vivo and in vitro in references 17 and 26.
Comments on the Quality of English Language
.
Reviewer 3 Report
Comments and Suggestions for Authors
Title
It is informative but it is not exhaustive: please add the type of aquafeed employed (woody plant-based feeds)
Abstract
Line 17-18: please, include a comment about differences or not among the different 8 groups
Line 23-25: please, remove the last sentence because it is not an acquisition of the current study and growth of tilapia is not reported.
Introduction
Line 50-: the references 9 and 10 are not responding to the fish species reported in the text
Line 62-: please, introduce a description of the benefits of the use of woody plants and their innovation in aquafeed with specific details of the Moringa oleifera, Neolamarckia cadamba and Broussonetia papyrifera and add references.
Materials and methods
Line 73: the reference is not numbered and maybe corresponding to reference 18
Line 81-84: although there is a remind to the supplement data, I suggest to add if the feeds are isoprotein and isoenergetic
Line 91: please add the initial mean body weight of the tilapia employed for the trial
Line 105-110: please, report how many fish were collected for the gut sampling (2?) and intestinal sampling (4?)
Results
Line 194-196: it is not correct to say “the four probiotic groups increased….” because no different letter are reported in Tab. 1 concerning VW
Line 217-218: this statement has no respondence in Table 2 because no significant difference is reported
Figures
Figure 2: the first chart is without letter A in the legend
Tables
Table 1: the legend is not reported. Please, add P value
Table 2: in T-AOC where is letter b alone? Please, add P value so it is possible to show if MDA is significantly different among the 8 groups
Discussion
Line 342-: please, give a discussion of the results also considering the woody plants used because not only differences were showed between probiotic and not probiotic groups. If this case, is found please stress that three different woody plant-based feeds did not affect gut and intestine.
Author Response
Please see the attachmen
